# An Overview on Railway Impacts on Coastal Environment and Beach Tourism in Sicily (Italy)

Irene Cinelli [1], Giorgio Anfuso [2,*], Sandro Privitera [3] and Enzo Pranzini [1]

[1]  Dipartimento di Scienze della Terra, Università di Firenze, Via Micheli 6, 50121 Firenze, Italy; irene.cinelli@hotmail.it (I.C.); enzo.pranzini@unifi.it (E.P.)
[2]  Departamento de Ciencias de la Tierra, Facultad de Ciencias del Mar y Ambientales, Universidad de Cádiz, Polígono Río San Pedro s/n, 11510 Puerto Real, Spain
[3]  Centre for the Conservation and Management of Nature and Agroecosystems, University of Catania, CUTGANA, Via Santa Sofia 98, 95123 Catania, Italy; sandro.privitera@unict.it
*  Correspondence: giorgio.anfuso@uca.es

**Abstract:** The main aim of this paper is to analyze the development of the railway network in Sicily (Italy), where it runs close to the sea on two of the three sides of the island, and give an overview of the related impacts on coastal environment and tourism. In order to achieve such an objective, the impacts of the railway network were analyzed according to coastal typology (distinguishing between rocky and sandy coastal sectors) and distance from the shoreline (dividing distance values in concrete intervals). Rails were mostly emplaced in flat coastal areas due to the island's rugged terrain: out of 1592 km of railway, ca. 350 km is located less than 1000 m from the shoreline (123 km on rocky sectors and 227 km on beaches and coastal plains). On sandy beaches and low sandy coastal sectors, approximately 6 km of track is within 25 m from the shoreline, a value rising to 16 km if a 50 m distance is considered, 48 km at 100 m and 103 km at 200 m distance. In correspondence of rocky platforms and high cliffed sectors, data reported for short distances between the rail and the shore are similar to ones observed along sandy coastal sectors, but differ when distance increases, i.e., there is only 32 and 47 km of railway respectively within 100 and 200 m from the shoreline. The emplacement of the railway embankment on beaches and dunes favored coastal squeeze and enhanced coastal erosion due to wave reflection on the embankment, which had to be protected by hard structures. Impacts on rocky sectors, with respect to beach and dune systems, are generally low because such sectors are usually stable (they do not need to be protected), less attractive to tourists and present small urban development. Tourism was affected by reducing landscape quality, beach access and width. More detailed studies and monitoring programs are necessary to locally assess the detailed impacts of the railway network, with this study constituting a preliminary but useful approximation to determine which coastal sectors are potentially the most affected. Results obtained in this paper can stimulate similar researches in other countries to prevent or decrease railway impacts on "Sun, Sea and Sand" tourism and, in general, on the coastal environment.

**Keywords:** beach access; coastal erosion; coastal squeeze; littoralization; coastal management; shore protection



## 1. Introduction

Railways have been the engine of economic and social progress in many countries and a symbol of progress itself. This is even more true for coastal areas, since the railway network was expanding there at the same time in which people were becoming aware of open-air activity healthiness and curative quality of sea air, an awareness that supported the growth of the summer holidays phenomenon.

Following Corbin [1], the recreational use of the coastal environment is a relatively recent trend since people lost their ancestral fear and terror of beaches in the 17th century, and only since the 20th century has a wider part of the European population reached

an economic level allowing them to travel and enjoy holidays far from their homes, and beaches started to be considered places of rest and relaxation [2]. The use of beaches for leisure transformed coastal areas into places of strong economic interest and highly productive spaces [3].

Watering places attracted the railway, and the railway stimulated their development in an almost uncontrolled positive feedback process, whose negative consequences manifested a few years later, although, in some cases, the local population already opposed the construction of the railway, but always without success [4].

At many places, the emplacement of the railway along the coast was a necessity since land morphology is one of the most important factors limiting railway development. In order to have a milder slope route, in mountainous or hilly regions, tracks are frequently placed along the coast, sometimes on the beach itself. This was the fastest and less expensive solution, especially until the beginning of the 20th century, when excavation of tunnels was still exclusively carried out with rudimentary tools and the massive use of explosive charges. The election of such emplacement was also linked to the uncertainties about the ownership of the land located between the shoreline and the limits drawn on the cadastral maps in a constantly evolving environment, and this favored aggressive railway companies in any legal disputes [4].

The election of the "alongshore solution" characterized many projects, even when originally the railway had to ultimately reach inland destinations, e.g., the Tokaido railway, linking Tokyo and Kyoto/Osaka, the first railway line in Japan. Its construction started in 1883 but, in 1886, the route was changed and moved to the coast [5]. Often, since the first phases of construction, stability problems of the railway embankment emerged and continued during the following years and decades [6]. As observed by Grant and O'Callaghan [7] south of Dublin (Ireland), between Killiney and Bray, shoreline erosion produced several realignments of the main railway line since its construction in 1856, but only one kilometer of the track was later defended by a seawall in the 1884–1886 period. Other cases are observed along the Black Sea, where the railway line runs very close to the shore, and in Georgia, where several parts of the railway were completely destroyed twice or even three times. At many such places, it has not been possible to relocate the railway further inland and the construction of tunnels was required at two sites [8].

In Russia, out of the 79.4 km of the Black Sea railway from Tuapse to Sochi, only the urban route and a segment of 600 m are not on the beach or at the foot of a cliff. Huge coastal protection works have been carried out since its construction and a fiberglass panel solution was also adopted to make up a seawall [9].

Railway relocation is extremely expensive at present demolition/construction costs and, even where the land is owned by the railway company, such a project is not carried out, e.g., in New Zealand where, although the Main South Line between Dunedin and Timaru is severely affected by storms, the KiwiRail company have no plans of moving it inland [10]. It is worth noting that in 1927, the Railways Department *started to advertise its lines to the coast with attractive posters urging New Zealanders to "Follow the Sun" and enjoy the scenic and therapeutic charms of destinations like the Bay of Islands, Tauranga, [...] and Timaru—to "Best Reached by Rail"* [11].

Environmental problems were not only related to railway proximity to the shoreline, but also to the fact that, quite often, sediments from the beach were mined to build up the embankment, e.g., in Georgia where, in the 1920s, railway construction staff mined beaches for pebbles and, to protect the line, seawalls and breakwaters were emplaced [12], and in the Liguria region (Italy), where beach sediments were used for the same purpose in the 1860–1870 period, thus constituting the first reason for shoreline retreat on the western side of this region [6].

Despite the presence of protection structures, railways are still exposed to storm waves in many countries, e.g., in the UK, 150 km of the operating railway network in Wales is on the coast [4] and, in January 2014, the track was damaged at several places such as Machynlleth, Barmouth and Pwllheli, which took five months to repair. At Dawlish,

Devon, the line was closed for two months because the protecting seawall was damaged in February 2014 and tracks were left hanging in mid-air [13]. During the January 2014 stormy period, the Cambrian Line along the west coast of Wales, from Machynlleth through Barmouth to Pwllheli, was also breached in several places and stranded trains had to be taken away by road on low-loaders [14]. In Ireland, more than 100 km of the railway between Dublin and Wicklow needs new protection from wave attacks [15]. In South Africa, the South Coast railway line experienced severe damage in 2007 at Mtwalume and Sezela due to the combined effects of storms in coincidence with equinoctial high tides [16].

Sea-level rise impacts on transport infrastructures have been highlighted by Dawson et al. [17] for England, but similar warnings arrive from different countries, e.g., from Morocco [18] to Australia [19]. Long-term plans to face this problem are carried on in many countries, like in Ireland [15], and in Australia, where only 20% of the 932 km of the total railway line length is within 5 km from the coastline—but never too close to it [19]. Adaptation projects have been developed in India [20] and The Netherlands too [21].

The coastal environment is severely impacted by the emplacement of the railway line too close to the shoreline and the construction of defense structures to protect it, usually revetments and seawalls. Therefore, the coastal environment is often affected by beach erosion problems, the impossibility of landward migration of coastal ecosystems (or coastal squeeze, [22,23]) and landscape degradation because of the presence of railway embankments and defense structures that makes the coast unattractive [24]. Concerning coastal tourism, it is severely hampered by beach surface reduction, access and use limitations and decrease in landscape attractiveness, which is one of the most important aspects of beach selection [25,26]. Considering all the above aspects, it is therefore useful to reconsider the oppositions that, at places, the local population made in the past to the emplacement of the railway tracks close to the shoreline and reflect on how motivated such resistances were. Evidence of the railway line impact on coastal environment and associated tourism gives useful information to decision-makers to implement sound management actions to slow down such impacts.

In Sicily (Italy), the railway runs close to the sea on two of the three sides of the island perimeter, connecting the major towns located at the coast, i.e., Palermo, Messina, Catania and Siracusa (Figure 1). Even though in this Mediterranean island tourism arrived later than in other watering sites, like southern Britain, Normandy, French Riviera and northern Italy and Spain, today the "3S" tourism (Sun, Sea and Sand, [27]) is one of the main economic activities.

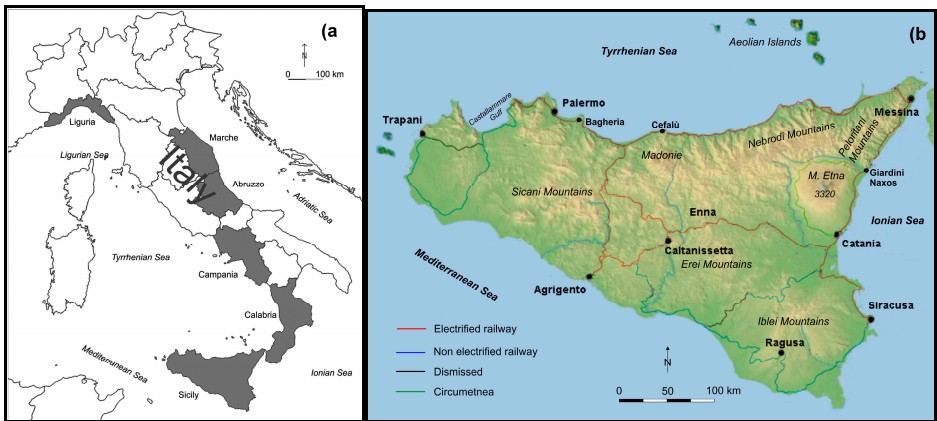

**Figure 1.** (**a**) Location map of Italy with regions later mentioned in the text; (**b**) Railway network in Sicily. Source: Base map obtained from Wikipedia and Railway network from Google Erath.

The main aim of this paper is to determine, at a regional scale (i.e., Sicily), coastal areas potentially affected by the railway emplacement [28] and describe from a theoretical point of view (but giving concrete examples too) the related impacts on "3S" tourism and, in general, on coastal landscape and ecosystem. To achieve such objectives, three main

aspects of the railway emplacement are considered of relevance and therefore analyzed, i.e., distance from the shoreline [4,8,17,19], location with respect to coastal settlements and coastal typology [22,23]. Regarding the former, it is assumed that the smallest is the railway distance from the shoreline the highest is its impact since the railway (i) is directly (or very closely) emplaced on the coastal environment destroying it or severely limiting its landward migration (thus favoring coastal squeeze [22]) and (ii) it can be easily affected by erosion processes and therefore coastal protection structures may be needed to protect the embankment producing further impacts on the environment. Regarding the latter, sandy coastal systems, which include sandy beach and dunes, are considered more vulnerable with respect to rocky sectors, which include rocky shore platforms and cliffed sectors, because sandy coasts are very sensitive to erosion processes, present a great ecological (e.g., if dune systems are present) and tourist-related value. Therefore the method used in this paper allows us to give an overview at a regional scale on railway emplacement and potentially impacted areas but further site-specific studies are afterward required to investigate particular locations. Results from this study can influence future projects of managed retreat to mitigate the negative effects that the railway has caused along the coast. At the same time, this paper can stimulate similar research in other countries where the impact of the railway on the coastal environment and on "3S" tourism has not been fully analyzed and/or serve as an advertisement for other countries to avoid problems trigged by the railway construction close to the shoreline.

## 2. The Italian Railway Framework

In Italy, as observed for other countries, railway construction has been accompanied, according to a cause–effect mechanism, by considerable socio-economic and territorial transformations. Railway had positive effects on the development of seaside tourism, similar to Liguria with the connection with the French Riviera but, at many long sectors connecting the north with the south of the country, the morphology of the peninsula's terrain, with the Apennine ridge that often extends to the sea, has conditioned the train tracks emplacement, and again, like in Liguria, they were located too close to the shoreline (Figures 1 and 2).

As a negative consequence, many beaches in Liguria have disappeared both for the previously mentioned extraction of sand and gravel, both for the wave reflection on the railway embankment [6], a process that required the construction of revetments, detached breakwaters and groins, with a progressive armoring of the coast. A similar process occurred on the Adriatic coast, where the railway runs very close to the sea along tens of kilometers [29]. There, opposite to the Liguria case, at the time of the railway construction, the strip of land close to the shore was almost uninhabited, with existing settlements safely located on the hills facing the sea.

The railway attracted the population and induced the establishment of satellite settlements that acquired more and more importance in following years, frequently determining the abandonment of the pre-existing villages safely located on the hills facing the sea: toponyms as "Scalo" (freight yard) and "Stazione" (station) were quite common among new settlements and, later, the word "Marina" was added to indicate a further shift to the shore [30]. Exemplary is also the case of Calabria where the first railway line was constructed in the 1866–1881 period along the Ionian side, an easy task linked to the favorable morphological characteristics. After that, during the 1881–1895 period, the Tyrrhenian line was emplaced along a predominantly rocky coast. Both coastal railway lines became a formidable instrument of observation of uncontaminated nature for the Calabrian people themselves: a crown of hill town centres overlooking rocky ridges and empty beaches [31]. The railway in Calabria attracted the population to the coast pushing the development of coastal tourism. Today, the entire 736 km-long coastal perimeter of Calabria is bordered by the railway, which lives out only a few small headlands: that element of initial development constituted in the following years an obstacle to the tourist offer and asked paramount investment in coastal protection projects, e.g., the largest single beach nourishment project

with quarried aggregates in Italy (1.1 Mm$^3$), within 19 "T" shaped groins, was carried out by the National Railway Company to protect the 6 km-long San Lucido–Paola railway segment [32].

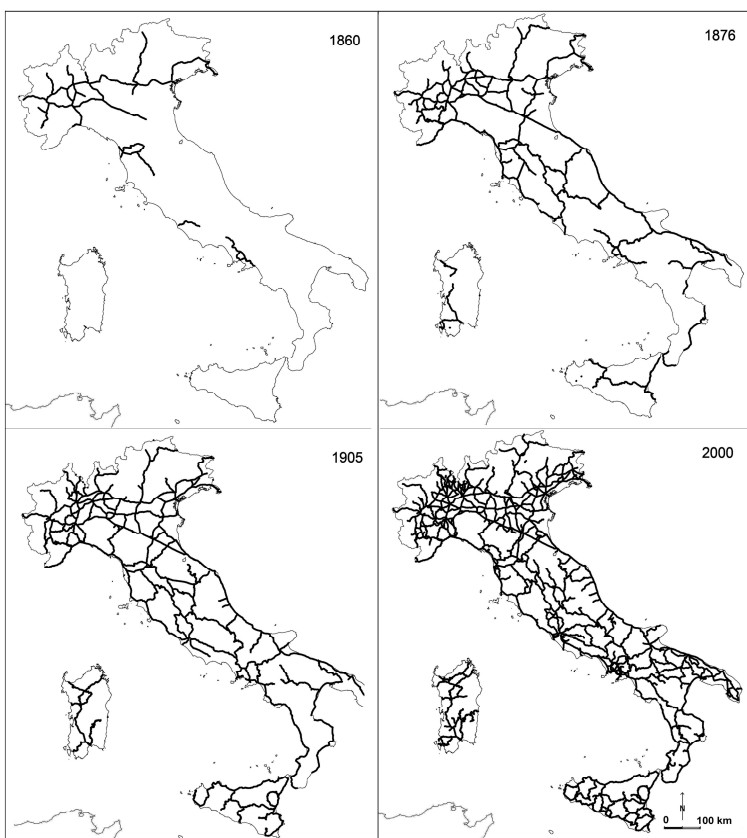

**Figure 2.** Railway network evolution in Italy. Source: authors.

Railway location was usually decided by National Railway Company managers, far from local interests and operating according to a cost/benefit principle that did not take into account the residents' will and economic necessities along the route. The railway location often constituted a strong limitation on local tourism development and an obstacle to the implementation of sound coastal management actions. As a result, the response of populations to railway construction was usually contradictory as in the case of Sanremo on 25 January 1872, when the first train arrived from the French Riviera, bringing rich French and English tourists, but the first beneficiaries, the hotel owners, did not like the longshore project motivated by the need to have tracks entering the harbor (Actually the very first run was in the opposite direction!). The same was observed at many places in Catania, where people opposed the railway reaching the coast to connect the harbor: the train cut the town into two parts, constituting a barrier separating the town from (and limiting the view of) the sea, all in the name of progress. Therefore, even where the beach was not lost, the barrier constituted by the railway embankment and the danger/prohibition of crossing the tracks allowed the access to the beach only at very limited points, through level crossings or underpasses–usually narrow tunnels built more for trickles pass and not to create comfortable pedestrian access to the beach.

## 3. Study Area: Morphology and Railway Setting

Sicily has a total population of ca. 5 million inhabitants with Palermo as the most important town, followed by Catania, Messina and Siracusa (Figure 1). It is the largest island in the Mediterranean Sea with a surface of 25,468 km$^2$ or 27,708 km$^2$ when the minor islands are also considered. The coastline, 1484 km-long, is a microtidal environment with

a spring tidal range of 0.6 m, constituted by a mostly steep and rocky northern coast, a sandy southern coast, and a very heterogeneous eastern coast. All such aspects constitute the base of a tourism-based regional economy [33].

Concerning Sicily's territory, 25% consists of mountains, especially located in the northern part of the island, 61% and 14% respectively consist of hills and plains, mostly observed in the southern part. Such complex orography has resulted from the Alpine collision (Late Mesozoic–current Cenozoic), which gave rise to the mountainous belt formed along the Africa–Europe plate boundary that links the African Maghrebides to the west and southwest, with the Calabria and the Apennines to the east and northeast [34,35]. This activity resulted in the formation of various mountainous chains, belonging to the Italian Apennines: (i) the Sicani Mountains, in the central-western part of the island, reach a maximum height of ca. 1600 m; (ii) the Madonie (with a maximum height of ca. 2000 m) and (iii) the Nebrodi (ca. 1800 m) chains run parallel (and close) to the northern coast of the island, facing the Tyrrhenian Sea, and (iv) the Peloritani Mountains (ca. 1400 m) run close to the Tyrrhenian and the Ionian coasts respectively on the northwest and northeast part of the island (Figure 1). The Etna mountain, the highest (ca. 3300 m) active volcano in Europe, is located on the eastern coast of the island on the Ionian Sea. Additionally, the southern part of the island is shaped in a smooth and wide plateau, geologically belonging to the African plate.

Eustatic and tectonic terraces are extensively present along the Sicily coast, including many assigned to the MIS 5.5–Tyrrhenian, allowing to identify areas of rapid uplift in the east end of the island (up to +175 m a.s.l.), slower uplift in the north one (+29 m), and relative stability in the northwest one (from +2 to +18 m) [36–39].

The railway network development i] Sicily started in the middle of the 19th century, to serve the growing sulphur mining that demanded extraction sites be connected to harbors. When this activity declined, the railway was used to deliver oranges and lemons to the northern Italian and European markets and was only at the beginning of the 20th century that it acquired a certain tourist use. However, several lines were dismissed from the 1960s to 2010 for a total length of 700 km, at the beginning as a consequence of the reduction of sulphur production and, later, because of the competition of road transport. Regarding the railway construction phases, after several aborted projects lead by different financing groups, the first Sicilian railway segment, namely the ca. 30 km-long Palerm–Bagheria line, was inaugurated on 28th April, 1863 (Figure 1). Only in 1895 did a 224-km-long railway segment, i.e., the "Tyrrhenian" one, connected Palermo with Messina. It runs almost entirely along the coast because of the presence of the northern Sicilian chains, i.e., the Peloritani, Nebrodi, Madonie and Sicani mountains (Figures 1, 3 and 4).

This new railway segment was connected with the existing Messina–Catania one (built between 1866 and 1871) that, because of the Peloritani Mountains, was mainly emplaced very close to the coast (47 out of 96 km), especially between the city of Messina and the town of Giardini-Naxos (Figures 1 and 4). This location was also possible because the low level of coastal urbanization that started in Sicily in the 1960s, with the "economic boom", reflected by large financial investment in private construction.

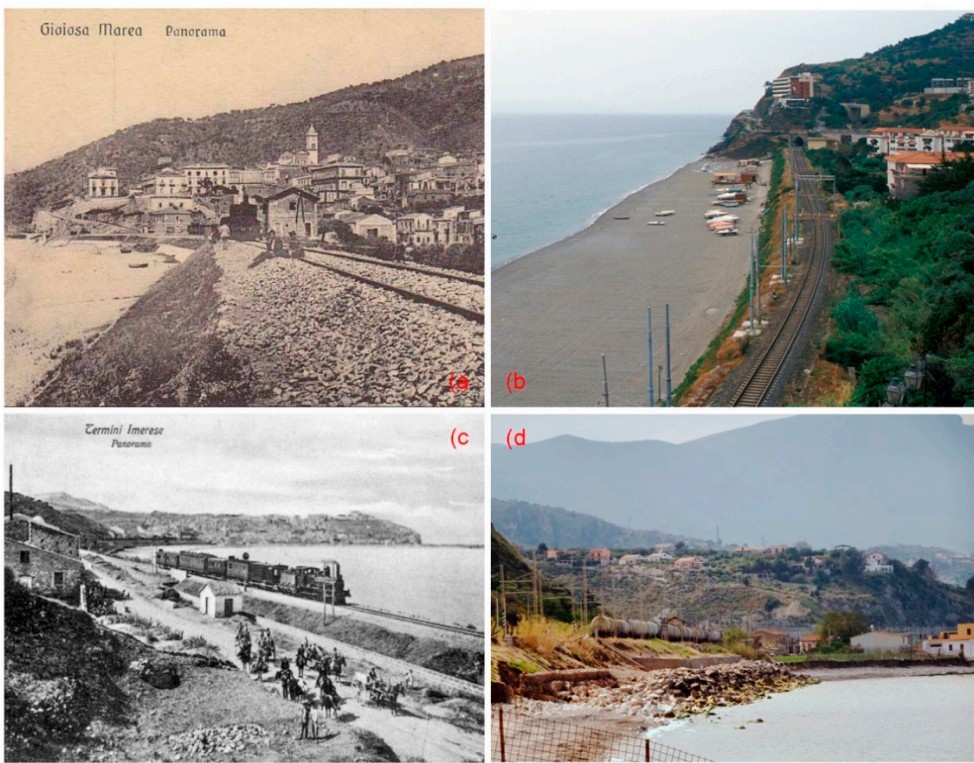

**Figure 3.** Example of railway segments running close to the sea along the Palermo–Messina line. Postcard portraying the railway at Gioiosa Marea at the end of the 19th century (**a**) and now protected by a nourishment project carried out in 2007 (**b**). The railway at Termini Imerese depicted by a postcard at the end of the 19th century (**c**) and now protected by a riprap (**d**). Source: authors.

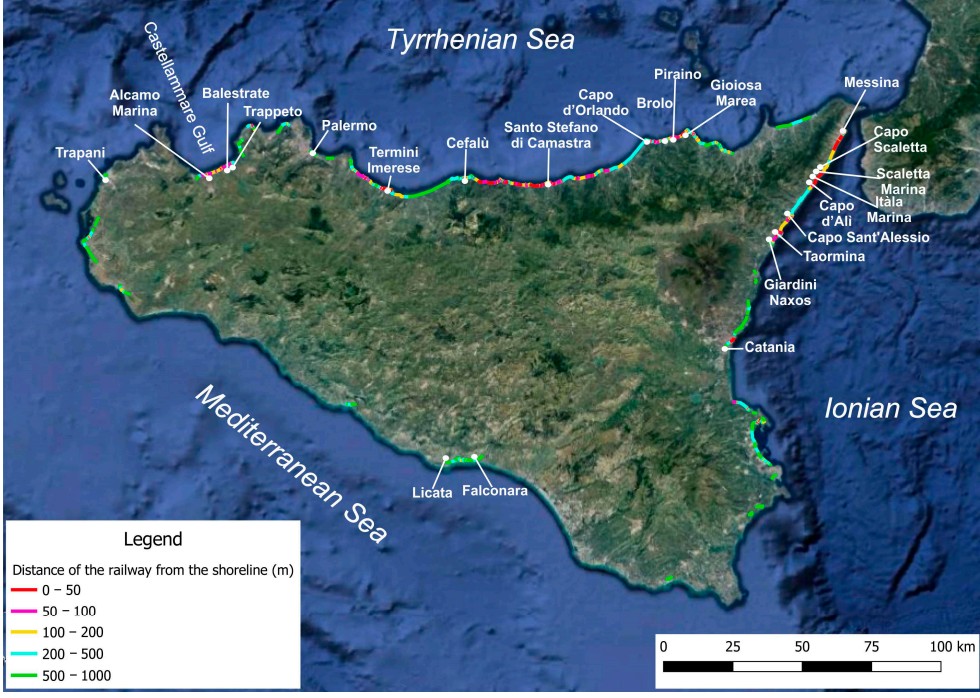

**Figure 4.** Railway segments distribution according to their distance from the shoreline. Source: base image from Google Earth.

Many coastal railway segments run on the 1st order marine terraces that extensively surround the Sicilian coast [36–39]. From Licata to the Falconara, the railway was built

partly on the alluvial terrace of the Salso River and partly on the internal limit of a 1st order marine terrace at ca. 25–30 m a.m.s.l. For this reason, it runs away from the sea, making wide curves to follow the morphology of the valleys in correspondence with the main streams (Figures 4 and 5). This is because the railway cannot run slopes steeper than 3% and a straight path would have involved the construction of long bridges. On the northern coast, along the ca. 75 km segment between Capo d'Orlando and Cefalù (Figure 4), the railway runs parallel to the shore, lying on the Tyrrhenian terrace gradually rising from ca. 10 m a.m.s.l. at Capo d'Orlando to ca. 25 m at Cefalù. The same is observed at Castellammare Gulf (Figure 4), where the route partly runs on a Tyrrhenian terrace at ca. 6–12 m a.m.s.l.

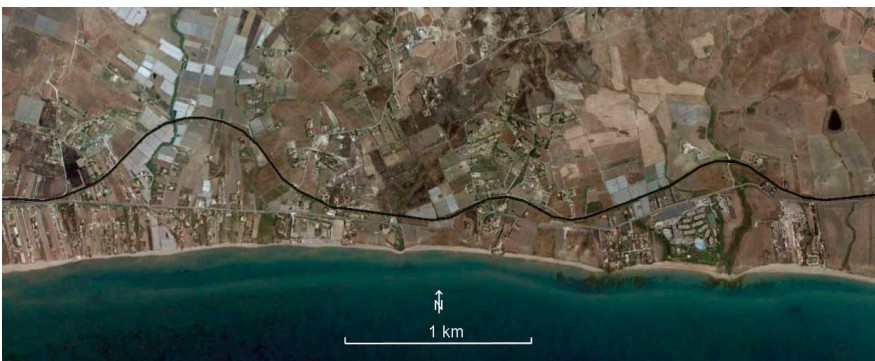

**Figure 5.** The railway east of Licata, running on a marine terrace, turns inland to cross the outlet of three small valleys. Source: base image from Google Earth.

## 4. Methodology

Investigations were carried out to analyze the emplacement of railway segments running inside a 1-km-wide coastal belt, which were later classified according to their mean distance from the shoreline following the rationale that the smallest is the distance between the railway and the shoreline the highest is the railway impact on coastal environment and related ecological and tourist values. Therefore the information presented in Figures 4 and 6 constitute a preliminary approach to determine the most sensitive coastal sectors in Sicily. Distances were measured on Google Earth images imported in GIS and zoomed to an approximate scale of 1:200. Due to the limited tidal range of coastal environments in Sicily, the measurement accuracy was estimated to be around 3 m. The use of aerial photographs and satellite images in environmental coastal studies is a wide and recognized practice [30,36,40]. The mean distance value of railway tracks from the shoreline was averaged for segments ranging in length from 50 to 2000 m, depending on the morphology of the coast: longer segments were established where the coast is straight and the railway runs parallel to it, and shorter ones on rocky indented coast or where the railway has an oblique direction.

Railway distance from the shoreline was analyzed for different coastal morphologies, sandy and rocky coast, for which cumulative curves and histograms for 100-m-wide classes were produced. Sandy coasts were considered more vulnerable to erosion processes and coastal squeeze and of greater tourist and ecological (in the case of the existence of well-vegetated and developed dune ridges) value than rocky ones. Further, railway position with respect to settlements was analyzed in correspondence with railway segments closer than 100 m from the shoreline in sandy sectors (48 km of railway). In such cases, the analysis, based on 25 m-wide classes arbitrarily established, aimed at assessing whether the railway separates residential buildings from the sea, and identified whether reciprocal posit is influenced by the age of the settlement. For those sectors, the length of the coastline protected by artificial structures was measured. Seawalls, rip-raps and detached breakwaters were considered with their longshore length, whereas for groin fields, the distance between the first and the last element was considered. Field inspections were carried out to check the accuracy of results.

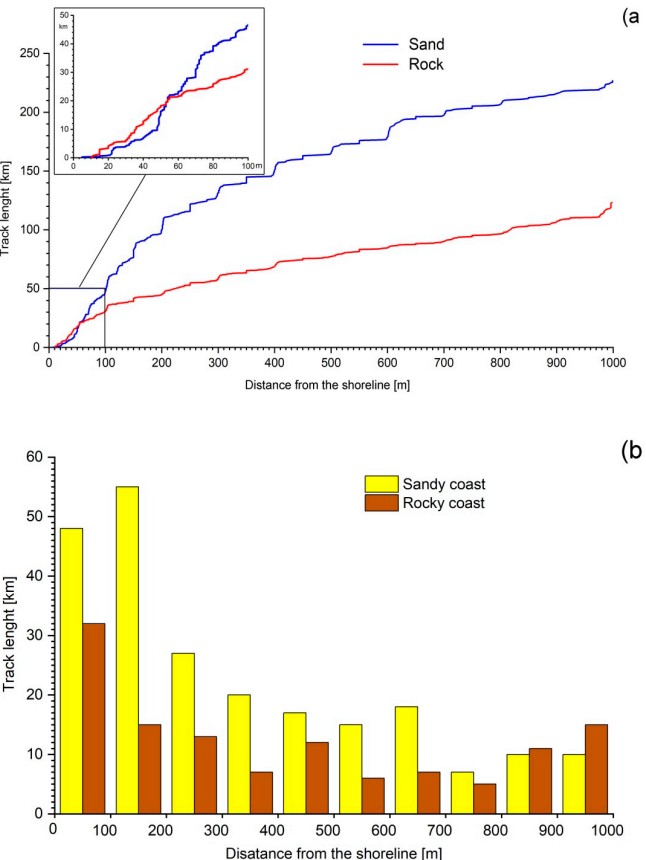

**Figure 6.** Railway segments distance from the shoreline according to coastline typology: Cumulative curves (**a**) and histograms for 100-m-wide classes (**b**). Source: authors.

## 5. Results: Distribution of Railway Segments

Approximately 350 km out of the total 1592 km island's railway length is located at less than 1000 m from the shoreline (Figure 4): 123 km runs on rocky coastal sectors and 227 km on beaches and coastal plains (Figure 6a). Although the railway position is conditioned by several different factors (coastal morphology, cliff stability, settlement distribution, etc.), such data clearly reflected the necessity of the railway to be emplaced in flat coastal areas due to the complex orographic configuration of the island. However, a more careful reading reveals further interesting aspects.

In correspondence of sandy beaches and low coastal sectors, which are very sensitive to coastal erosion and present a great tourist relevance, approximately 3.8 km of track is within a distance of 25 m from the shoreline, a value rising to 16.4 km if a 50 m distance is considered, 48.9 km at a 100 m and 100.1 km at a 200 m distance (Figures 4 and 6a,b). Some railway segments are particularly close to the sea, e.g., on the 30.8 km-long Cefalù–Santo Stefano di Camastra line, 10.3 km has a mean distance of 32 m from the shoreline (2.9 km are closer than 25 m), and on the 47.4 km-long Messina-Taormina line, the mean distance from the sea is 29 m (4.8 km is closer than 25 m, Figure 4).

As far as rocky and high coastal sectors are concerned, data are similar to sandy coast for short distances (5.5 km of railway runs inside a 25 m distance from the shoreline and 18.1 km inside a 50 m distance), but there is only 31.1 and 44.8 km of railway respectively within 100 and 200 m distances from the shoreline (Figure 6a,b).

Considering the whole railway network, the nearest segments to the sea are observed on the rocky coast, where rails were located just a few meters from the shoreline but at a height sufficient to stop them from being reached by waves, at least on less exposed coastal sectors. The figure showing cumulative distances (Figure 6a) evidences that, within 60 m from the shoreline, there are more kilometers of the railway on high coastal sectors than on

low ones, and that the reduction of interest in placing the railway close to the sea is evident where the island perimetric route was abandoned, especially at distances higher than 800 m. The 100 m-wide distance classes' histogram (Figure 6b) shows that once the coast is left, there is no preferred position to host the railway. On the contrary, when the railway runs on flat areas, a wider strip is favorable for its location and the cumulative curve rises more gradually and, on the histogram of Figure 6b, the modal class is the 100–200 m one.

While in Liguria the railway had to adapt to the pre-existing settlements, at least respecting traditional access points to the sea, e.g., by means of frequent and wide passages, in general, in the rest of Italy, the railway was constructed before the migration of population from the hinterland to the coast—which started between the two World Wars and, in Sicily, a bit later. This allowed the railway to run freely along the coast without finding any obstacle for long distances, as in Sicily where there is 48.8 km of rails closer than 100 m from the shoreline. Buildings had to adapt themselves to the pre-existing railway position (Figure 7). The windows of houses and hotels face directly onto the tracks and the rare accesses to the beach take place through narrow underpasses, often built where the railway had to cross streams (Figure 7).

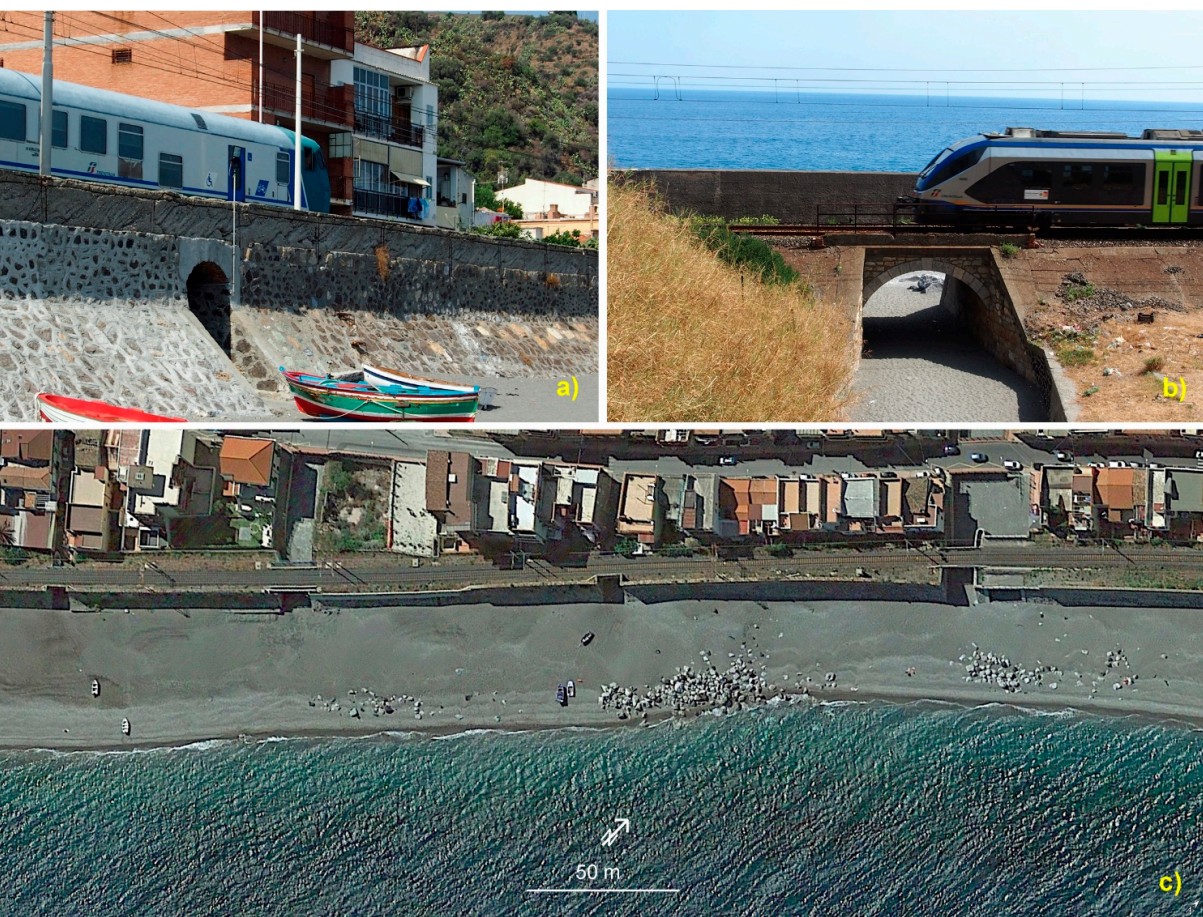

**Figure 7.** (**a**,**b**) land and sea-side view of one of the railway underpasses at Itàla Marina; (**c**) access to the beach at Scaletta Marina is through narrow pedestrian underpasses or stream beds; scattered stones belonging to old shore protection structures are visible on the beach. Source: (**a**,**b**) authors; (**c**) Google Earth.

The above was reflected by the analysis of railway position in respect to the distance from the shoreline and the location of settlements. Analyzing railway segments running within a 50 m distance from the shoreline (16.2 km), it was observed as only 2.1 km of railway tracks had to be emplaced landward of the pre-existing houses, whereas settlements were separated by the sea along a total railway length of 14.1 km (Figure 8). Considering railway length running in a coastal strip between 50 and 100 m from the shoreline (32.6 km),

houses appeared on the seaside of the railway along a length of only 22.0 km and beach access is allowed through rare underground passages often made more for crossing streams than for the passage of people (Figure 7). On the contrary, the railway runs on the inland side of old towns and villages (e.g., Catania, Cefalù, Palermo, Siracusa, Gela and Licata), which were built centuries before the railway construction.

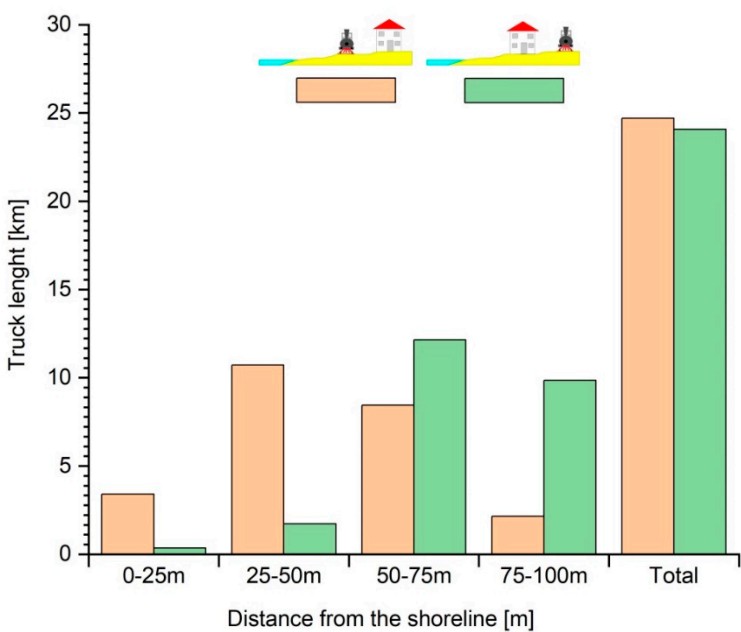

**Figure 8.** Length of the railway segments differently positioned in respect to the beach/settlements. Source: authors.

It was analyzed the length of railway sectors protected by hard engineering structures on sandy coastal areas. Railway sectors within a 50 m distance from the shoreline are protected by hard engineering structures along 8.0 km, i.e., 27.3% of the whole length of the railway sectors within such interval. Percentage of protected railway sectors between 50 and 100 m from the shoreline reduces to 9.7% (Table 1).

**Table 1.** Length of railway defended by coastal protection structures.

| Distance from the Shoreline (m) | Length of Railway Segments Located between the Shoreline and Settlements | Length of the Segments Defended by Artificial Structures | Percentage of the Railway Length Defended by Artificial Structures in Each Coastal Strip |
|---|---|---|---|
| 0–50 | 29.2 km | 8.0 km | 27.3% |
| 50–100 | 17.2 km | 1.7 km | 9.7% |
| 0–100 | 46.4 km | 9.6 km | 20.8% |

## 6. Discussion

### 6.1. Railway Line and Coastal Squeeze

Coastal zones are affected worldwide by a process defined as coastal squeeze [22,40,41]. This consists of the reduction of the coastal environment linked to both natural processes (i.e., coastal erosion and flooding, related to an increase in storminess and present and future sea-level rise trend [42]) and human interventions (e.g., land-use change reflected by the emplacement of infrastructures/settlements or the transformation of a natural area into agriculture devoted one), as well as in the compression of the coastal environment because of the impossibility of landward migration, e.g., an eroding beach or a tidal flat limited landward by a seawall [41,43]. As a result of the coastal squeeze, coastal fragmentation and changes in natural processes are observed and coastal ecosystems no longer have the

necessary conditions to maintain their essential functions, suffering degradation and loss of biodiversity [42,44].

In Sicily, coastal occupation due to the construction of summer houses and expansion of coastal towns and villages—rarely planned and frequently unauthorized [45]—constituted the major responsible for the recent and ongoing coastal squeeze [46] to which the railway contributed in different ways. At the end of the 19th century, the railway embankment already formed a few-meter-high fence along two of the three sides of the island (Figure 4), often placed on the beach (Figure 3). Tracks were emplaced directly on the coastal environment destroying valuable ecosystems, such as vegetated dunes [47,48], and contributing to coastal area fragmentation [49,50]. Even where a real embankment was not constructed, a sort of pebble revetment was placed to prevent sand erosion; this induced a floristic change from psammophilus (especially *Ammophila arenaria subsp. Australis* and, secondary, *Elytrigia juncea*, *Cyperus capitatus*, *Eryngium maritimum*, *Echinophora spinosa* and *Euphorbia paralias*) to stone-loving vegetation (e.g., *Crithmum maritimum* and *Limonium virgatum*). In many places, native vegetation was artificially substituted by eucalyptus, which arrived in Sicily at the beginning of the year 1900 and was soon used for dune fixation along railways [51].

Furthermore, one of the most important consequences of the "fence" effect was the impediment of coastal ecosystems' landward migration, especially of the beach system. This brought to the enhancement of beach loss. Since the beginning of the 1930s, but especially from the 1950s, several Railway Administration's technical reports [52], described as the tracks, defended by wide beaches during half a century, started to be seriously damaged by the waves, and coastal protection structures had to be constructed. Beach erosion, initially linked to the reduction of sediment inputs from rivers [53], further increased due to the emplacement of hard structures to protect the railway. Wave reflection on protection structures such as revetments or seawalls usually favors great seasonal sand volume variability with respect to the non-walled locations [54] and progressive beach profile lowering and erosion [55], also observed at other places [56]. As a result of erosion processes, the beach is progressively narrowed and no longer acts as a buffer by absorbing wave energy [57]. The loss of a healthy beach and dune system, which provide major reservoirs of sand, finally decreases natural coastal resilience, i.e., the intrinsic ability of the coast to accommodate changes induced by sea-level rise, extreme events and occasional human impacts, and therefore maintain the coastal system functions long term [58].

Because of the strong morphological constraints it had to adapt to, i.e., the presence of the Peloritani Mountains (Figures 1 and 4), and the coastal energetic conditions, the Messina–Catania railway has been the line that earlier and more intensively had to face coastal erosion. From Capo d'Alì to Capo Scaletta (Figure 4), serious problems started in 1933, and few groins were built after the railway was covered by sand during storms, but they were soon destroyed. Further railway interruption occurred in 1935, 1940, 1942, 1950 and 1951. To defend this coastal sector, in 1953, 13 groins were built south of Capo d'Alì; now, seawalls and groins protect the railway line along a total length of 6 km. South of Capo Scaletta the first damages occurred in 1942, and were repeated in 1945, 1946, 1947, 1949, 1950, 1951 [52]. In 1952, some groins were built after the railway was covered by sand during storms. The same groins were destroyed by waves' attack a few years later and after being rebuilt they suffered again serious damages in 1953 and 1954; in 1956, two breakwaters were emplaced in 1958. In the area of Itàla Marina (Figures 4 and 7), the first damages occurred in 1945, but similar events repeated in 1948, 1950, 1951, 1960, 1961 and 1962 [52]. At Fondaco Parrino (Figure 4), at the exit of the railway tunnel crossing Capo Sant' Alessio, between 1935 and 1938, a series of defense works were carried out to protect the railway. They consisted of a line of 30 large concrete blocks and a long bank sloped wall covered by limestone blocks that are still in good condition.

On the Palermo–Messina (Figures 1 and 4) line, problems arrived later, and only in October 1951 at Piraino was the embankment seriously damaged and was defended with gabions, but following a larger collapse in 1952, gabions were flanked with revetments.

Notwithstanding the continuous and expensive protection works, the railway in Sicily is still tremendously exposed to wave attacks as demonstrated, e.g., by the December 2019 closure of the Palermo–Messina line between Capo d'Orlando and Brolo, because of the erosion of over 100 m of embankment at the Brolo station. In these places, protection structures could be partially substituted or accompanied by nourishment works as carried out in other Italian regions, e.g., at San Lucido (Calabria), where the railway protection works implied a 2 million m$^3$ nourishment protected by "T" groins [30], thus providing a surface usable for bathing, although its access is quite limited by the tracks.

## 6.2. Railway Impact on Coastal Tourism

Both natural, e.g., landscape beauty and favorable weather climate and, especially, rich cultural heritage from the Greek to the Baroque period, make Sicily a very attractive touristic destination with ca. 5 million tourists recorded in 2019 [59]. The "3S" tourism sector achieves a great relevance too, especially due to local and national visitors, but the railway did not favor its development and related economic inputs as other areas of Italy by bringing great amounts of tourists, in this way partly balancing the negative impact of its infrastructure on the tourism itself. Only at the beginning of the 20th century the Messina–Catania railway line started to acquire certain relevance for tourist development, allowing the arrival of thousands of wealthy Italian and foreign tourists to Taormina, a place that during the *Belle Époque* became one of the most famous tourist destinations in Europe [60].

The emplacement of the railway line on the coastal environment affected tourism potentials reducing landscape quality, services availability and beach width and associated tourist attractiveness and use, aspects strictly linked to the recreation value of the beach that, according to Wilson and Liu [61] (p. 130), *got inordinate attention in the economic literature*. The coastal landscape is assessed by Williams [62] as one out of the five parameters of greatest importance to coastal tourists, being the others safety, facilities, water quality and litter. Scenery is a particularly appreciated resource for aesthetic, cultural, economic and historical reasons and local managers have to evaluate it in an objective and quantitative manner, as coastal *scenery is a resource, partly because of the economic value and partly because it is an accepted component of resource assessment programmes* [63] (p. 393). The most common coastal scenery classification is the "Coastal Scenic Evaluation System" (CSES) method proposed by Ergin et al. [64], which allows the assessment of coastal beauty in an objective and reliable way since it is based on the semi-quantitative evaluation of 26 natural and human parameters. Considering the 18 natural parameters of the CSES method, in many places, the railway embankment in Sicily had a huge negative impact on most of them since it was often directly emplaced (and therefore destroyed or severely damaged) on valuable morphological features such as the rocky shore platform, the beach, the dune system and the natural vegetation cover of the area (points 4–10 and 17 of the CSES method) and reduced the skyline landform view and vistas from the beach (points 12 and 15). Regarding the eight human parameters, the railway produced noise disturbance (point 19), made the built environment unattractive (point 23), occupied the buffer area between the beach and surrounding areas (point 24), and favored the presence of different unattractive elements on (or very close to) the beach as power lines, revetments, seawalls, etc. (point 26).

The railway location on the beach or dune ridges not only produced the loss of natural environmental and scenic qualities, but also impacted the realization of beachgoers' leisure activities and beach accessibility (Figure 7).

Beach and dune surface reduction was exacerbated by the enhancement of beach erosion problems linked to the construction of the embankment and protective hard structures on the beach. Therefore, at many places, the railway caused the loss of recreational beach activities, such as lying in the sun and playing games or the use of the beach as access to the sea. To solve such problems, especially beach erosion linked to hard protection structures [56,65], their use could be reduced and/or partially replaced by the execution of beach

nourishment and dune restoration works that would enhance beach width and associated tourist use [25,66] and environmental scenic beauty and ecosystem services [22,24,67].

Concerning beach accessibility (Figure 7), i.e., direct and easy access to the shore, it is strongly valued by beachgoers as observed by several researchers, e.g., [68], and it is considered an important aspect in the assessment of the overall beach quality [69].

Accessibility can be qualitatively assessed by the various litigations carried out to have this right recognized [70,71] or monetary calculated through the willingness to pay for extra passages [72]. *Il mare in gabbia* (The sea in cage) is a slogan firstly used in 1966 by one of the fathers of Italian environmentalism [73], to denounce the tourist settlement of northern Sardinia impeding to reach the sea.

The railway is not only directly affected the coastal environment and associated ecosystems and tourist uses since the barrier effect of the embankment influenced the real estate market and hotel prices in coastal towns and villages. Although no specific studies have been carried out on the economic impact on tourism of the railway barrier, the value of buildings and rooms in hotels with a sea view has been assessed by different authors. Benson et al. [74] (p. 68) in Washington found that an unobstructed ocean view adds 68.3% to value if the property is located very close to the water (0.1 miles). Concerning hotels, in Cyprus, sea-view room price is 11% higher than other rooms and similar values were recorded at ten Mediterranean tourist destinations [75]. In Veracruz, México, room prices increased by 8% and 57%, depending on the ocean view and accessibility to the beach, respectively [76] (p. 4). As far as train noise is concerned, in Oslo, Strand and Vågnes [77] observed that the doubling of the distance from the railway line, within a 100-m boundary, increased the property price by about 10%. In Sicily, the noise disturbance related to the railway likely has a stronger impact on house values, considering that because of the favorable climatic conditions especially during the summer period, people often keep windows opened at their homes.

## 7. Conclusions

In Sicily, because of the complex morphological and geological characteristics of the island, the railway embankment was mostly emplaced directly on the coastal environment, at a distance closer than 1000 m from the shoreline along 350 km out of the 1623 km that constitutes the island perimeter. Such a particular situation enhanced this investigation aimed at determining railway impacts on coastal environment and related tourism, by characterizing track emplacement according to the distance from the shoreline and coastal area typology. The rationale behind this investigation was that impacts are strictly related to:

(i)    Railway distance from the shoreline. When the railway is directly emplaced on (or it is very close to) the coastal environment, greatly affects it by damaging/destroying ecosystems and affecting tourist activities and impeding coastal landward migration;

(ii)    Railway position in respect to the settlements. If it is located between the houses and the beach, it has a higher landscape impact and limits beach access;

(iii)    Coastal typology. In general, sandy sectors (i.e., beaches and dunes systems), with respect to rocky sectors (rock shore platforms and cliffs), present greater ecological value (if large and well-vegetated dune ridges exist) and tourist interest. Further, sandy sectors are very sensitive to sea-level rise and storms impact that, under determinate conditions, can favor the landward migration of the whole coastal system, which is limited by the presence of the railway. Such problems are further enhanced when coastal protection structures (groins, breakwaters, rip-rap revetments, etc.) are constructed to protect the coast.

Therefore, within this paper, railway sectors located at a distance closer than 1000 m from the shoreline were further characterized. Approximately 123 km (out of 350 km) runs on rocky sectors and 227 km on sandy sectors and the latter needs particular attention from local managers. In such areas, the railway construction caused direct destruction and fragmentation of the environment, and the impossibility of its landward migration due to

the presence of the railway embankment that had often to be protected by hard defense structures, such as revetments and seawalls. The embankment had a relevant impact on the landscape too, made the skyline unattractive and favored the presence of numerous utilities and limited a relevant aspect for beachgoers, e.g., beach accessibility that, at many places, was allowed only through low and narrow underpasses. Further, hard defense structures emplaced to defend the embankment favored beach lowering and enhanced existing erosion trends. Associated beach surface decrease and dune ridges losses caused a diminution of tourist recreational activities and landscape beauty.

Beach nourishment and dune ridges restoration works could be carried out to enhance beach and dune natural values, coastal resilience and tourist attractiveness and use. A more complex solution to completely restore the coastal environment's natural and tourist functions consists in the inland relocation of the tracks and to leave the existing route management to the local administration that, as observed in the Liguria region, transformed it into a bike route passing inside the tunnels, which is greatly appreciated by tourists. In the Abruzzo region, coastal erosion necessitated rebuilding the railway alongshore line several times until it was relocated inland and, along the old course, a 60-km-long pedestrian/bike route was implemented, which is now part of the Ciclovia Adriatica (Adriatic Bike Route).

On a global scale, the adaptation of the railway network to actual environmental issues, i.e., the necessity of a substantial shift of freight transport from road to rail, mandatory to reach the decarburization levels established by international agendas, as well as the request for faster passenger connections, requires the creation of new routes. This study can stimulate similar research in other countries and increase the awareness regarding the railway impacts on "3S" tourism and, in general, on the coastal environment, and enhance the modification of existing tracks and/or the construction of new suitable segments, e.g., to define optimal routes in all those countries that, for morphological/orographic reasons, must place the tracks in the proximity of the coast.

**Author Contributions:** Data curation, I.C.; Formal analysis, E.P.; Investigation, S.P.; Writing—review & editing, G.A. All authors have read and agreed to the published version of the manuscript.

**Funding:** This research received no external funding.

**Institutional Review Board Statement:** Not applicable.

**Informed Consent Statement:** Not applicable. The study did not involve humans or animals.

**Data Availability Statement:** Data supporting reported results can be found asking directly of the first author.

**Acknowledgments:** This work as a contribution to the PAI Research Group RNM-328 of Andalucía (Spain) and the PROPLAYAS Network.

**Conflicts of Interest:** The authors declare no conflict of interest.

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
