# Peer review of "An Overview on Railway Impacts on Coastal Environment and Beach Tourism in Sicily (Italy)"

_sustainability, doi:10.3390/su13137068_

Round 1

Reviewer 1 Report

See reviewer remarks attached. 

Author Response

Response to Reviewer #1

First of all we want to thank very much the reviewer for very useful comments and suggestions. All new, corrected text is in blue.

Question 1:

 The Abstract contains too many historical or descriptive data, which is not its purpose. Authors should try to focus on the scope of their work, their methods of analysis and their findings and conclusions. All other facts may be well positioned in the other parts of the paper.

Answer:

We reduced such data and added new text concerning the focus and scope of the work.

Question 2:

Page 3, last paragraph – this is where the authors needed to justify the purpose of their work. At this present stage, what is presented needs more attention. Authors indicate they will analyze the emplacement of railways close to the sea, but do not clarify the scope and aspects of this analysis. The authors talk about economic activities and exploration of the way this impacted tourism in Sicily, but there are no economic analyses further in the text (the analyses are more around geographical developments, spatial analysis, etc.). As a main contribution, the paper seem to outline that the preliminary study it presents might stimulate future studies that will quantify how the tourism economic realities justify the presence of railways close to beach areas. It is also expected to stimulate similar research in other countries with similar tourism and railway network characteristics. None of these warrant solid research novelty and contribution. This is a lot more like directions for future studies rather than valuable research contributions. So, authors need to pay a lot of attention to justify the purpose of their work.

Answer:

We modified the text greatly enhancing the part devoted to explain the focus of the paper and the rationale behind it. We also added further text required by reviewer 2.

Question 3:

The aspects mentioned in remark 2 refer also to the Conclusions section. At this stage, it is a long summary of what has already been said, along with some more historical data and historical developments, yet the research novelties of the work cannot be properly traced. Authors need to work on this and improve their Conclusion section.

Answer:

We erased some information and added the relevance and rationale of this study.

Question 4:

Although the paper has good figures with good quality, some of those are more illustrations and do not bring that much of research value, while occupying a lot of space. For example, Figures 3, 8, 9 are images of several areas at different periods of time, which while visual do not improve so much the value of what is being described (which is well structured and clear). So authors should consider removing some figures.

Answer:

We eliminated figures 3 and 8 and reduced figure 9.

Question 5:

Section 4 – Methodology – what authors present as methodology, is a simple data collection accompanied with basic descriptive statistics analyses (in section 5). Authors could perhaps provide more specifics on their methodology, and also perhaps compare their approach with that of other similar studies. That will bring more value to their methodology approaches.

Answer:

We added new text explain the rationale of the work and the importance of aerial photos and satellite images in such studies at regional level.

Question 6:

On several occasions, there are small issues, e.g. “costal” instead of “coastal” (line 16), “trucks” instead of “tracks” (lines 378, 408,etc.)

Answer:

Thank you very much for your corrections, we amended all errors.

Question 7:

Line71 – authors need to clarify what they mean by “Huge defense work”, as it implies military activities not coastal protection activities.

Answer:

Thank you very much, we corrected it.

Reviewer 2 Report

The author's conclusions are convincing. The present study outlines the existing problems in a very persuasive manner and makes some very interesting suggestions. Altogether, the present paper is an very valuable scholarly approach.

Author Response

Dear Reviewer 

We want to thank you very much for your work and comments. We hope this paper will be of interest to scholars and coastal planners.

Best regards.

Reviewer 3 Report

The manuscript presents an interesting topic. Following are recommended to be considered for improvement:

  • The aim of the paper to be presented in the abstract as well
  • Figure 1 a) is not of the best quality, same for Figure 2 – if possible to provide a better quality
  • Please add the source for all Figures unless it is own calculation
  • What are the limitations of the research?
  • Please add references of other studies that used this methodology (if it is the case) and explain in the conclusion section how your work and others can stimulate further studies/ similar research as stated in lines 128-136?

Author Response

Dear Reviewer 

First of all we want to thank you for your work and comments that have been fully answered and amended in the new version of the manuscript. All new added text is in blue.

Specifically we have explained the main aim of the manuscript in the abstract and the interest of this study.

Figures 1 a and 2 have been improved, especially the contrast – as a result the limits and borders of land are much clear.

We added in figure captions the source and authorship of each figure.

Limitations of this research have been added at the end of introduction.

Unfortunately it was not possible to add new references since there are not much studies on this topic (i.e. the overall impacts of railway), and such studies have been already mentioned in the introduction and along other sessions too.

We added in the conclusions the required amendments. 

Best regards. 

Round 2

Reviewer 1 Report

In its revised version, this manuscript has been substantially improved. The authors have done many efforts to comply with the remarks of the reviewer.

There are several minor issues to address:

  • There is a keyword “ICZM” that is never used in the text, nor clarified. Perhaps an abbreviation is not the best keyword.
  • The paragraph starting at line 140 – this is the objectives of the paper and what the authors will do. Past tense is hence not quite relevant here.
  • Further on the paragraph starting at line 140 – there is this massive set of references “[4, 8, 17, 19, 22, 23, 25, 26]” that are there without evident reason, not to mention that they are too many. If those are mentioned for the first time, they will need further clarification (in their relevance and content). If they are already mentioned, perhaps they might be omitted.
  • Figure 1 – it is confusing to me that a figure both comes from a source and from the authors themselves. “Railway network in Sicily [28]. Source: authors. Further on this, authors need to clarify if they have obtained permission to reproduce figures from the source [28].
  • Line 598-599 – since this is the Conclusion, it would be better to use past tense to explain what the rationale of the study “was”, rather than what it “is”.
  • Lines 609-617 – there is a formatting issue in this bulleted text

Author Response

Dear Reviewer

Thank you a lot for your observations/corrections.

There are several minor issues to address:

  • There is a keyword “ICZM” that is never used in the text, nor clarified. Perhaps an abbreviation is not the best keyword.

 Thank you we changed it for "Coastal Management".

  • The paragraph starting at line 140 – this is the objectives of the paper and what the authors will do. Past tense is hence not quite relevant here.

Thank you we made the correction.

  • Further on the paragraph starting at line 140 – there is this massive set of references “[4, 8, 17, 19, 22, 23]” that are there without evident reason, not to mention that they are too many. If those are mentioned for the first time, they will need further clarification (in their relevance and content). If they are already mentioned, perhaps they might be omitted.

Thank you yes we reduced them and divided in two groups –one regarding railway distance from the shoreline and an other regarding coastal squeeze (it is related to coastal typology)

  • Figure 1 – it is confusing to me that a figure both comes from a source and from the authors themselves. “Railway network in Sicily [28]. Source: authors.

Yes we clarified that base map was obtained by WIKIPLEDIA and Railway network from Google Erath.

  • Further on this, authors need to clarify if they have obtained permission to reproduce figures from the source [28].

We erased such reference and we want to clarify that we used free access documents that are now mentioned in the figure caption.

  • Line 598-599 – since this is the Conclusion, it would be better to use past tense to explain what the rationale of the study “was”, rather than what it “is”.

Done.

  • Lines 609-617 – there is a formatting issue in this bulleted text 

Done.

Reviewer 3 Report

The manuscript was improved according to the recommendations.

Author Response

thank you very much for your work and efforts